# PointDMS: An Improved Deep Learning Neural Network via Multi-Feature Aggregation for Large-Scale Point Cloud Segmentation in Smart Applications of Urban Forestry Management

Jiang Li [ID] and Jinhao Liu *

School of Technology, Beijing Forestry University, Beijing 100086, China; jiangge@bjfu.edu.cn
* Correspondence: liujinhao@bjfu.edu.cn

**Abstract:** Background: The development of laser measurement techniques is of great significance in forestry monitoring and park management in smart cities. It provides many conveniences for improving landscape planning efficiency and strengthening digital construction. However, capturing 3D point clouds in large-scale landscape environments is a complex task that generates massive amounts of unstructured data with characteristics such as randomness, rotational invariance, sparsity, and serious barriers. Methods: To improve the processing efficiency of intelligent devices for massive point clouds, we propose a novel deep learning neural network based on a multi-feature aggregation strategy. This network is designed to divide 3D laser point clouds in complex large-scale scenarios. Firstly, we utilize multiple terrestrial laser sensors to collect a large amount of data in open scenes such as parks, streets, and forests in urban environments. These data are integrated into a practical database called DMSdataset, which contains different information variables, densities, and dimensions. Then, an automatic block integrated with a multi-feature extractor is constructed to pre-process the unstructured point cloud data and standardize the datasets. Finally, a novel semantic segmentation framework called PointDMS is designed using 3D convolutional deep networks. PointDMS achieves a better segmentation performance of point clouds with a lightweight parameter structure. Here, "D" stands for deep network, "M" stands for multi-feature, and "S" stands for segmentation. Results: Extensive experiments on self-built datasets show that the proposed PointDMS achieves similar or better performance in point cloud segmentation compared to other methods. The overall identification accuracy of the proposed model is up to 93.5%, which is a 14% increase. Particularly for living wood objects, the average identification accuracy is up to 88.7%, which is, at least, an 8.2% increase. These results effectively prove that PointDMS is beneficial for 3D point cloud processing, division, and mining applications in urban forest environments. It demonstrates good robustness and generalization.

**Keywords:** urban forest management; terrestrial laser mapping; deep learning neural network; point cloud semantic segmentation; forestry point cloud features



## 1. Introduction

More than half of the global population resides in urban areas. These cities necessitate extensive infrastructure and a wide range of services to support the densely concentrated population. These services include electricity grids, public and private transportation, water supply and sewage systems, telecommunication networks, healthcare facilities, banking services, educational institutions, childcare centers, nursing homes, social welfare programs, law enforcement agencies, and governmental operations. The operation of these services generates a vast amount of data. When collected and analyzed using state-of-the-art computer and information technologies, these cities transform into "smart cities"

that operate more efficiently, consume fewer resources, exhibit enhanced connectivity and security, and promote environmental sustainability.

There exists a comprehensive array of enabling technologies that can be implemented to facilitate the development of smart cities. These technologies include the Internet of Things (IoT) and sensing technologies, advanced telecommunication technologies like 5G, cyber-physical systems, cloud and service-oriented computing, smart grid systems, intelligent transportation systems, unmanned aerial vehicles, autonomous driving, machine learning, artificial intelligence, computer vision, and blockchain technology. The integration of multiple enabling technologies represents a valuable opportunity in the field of smart city research, and it is an emerging area of study.

By harnessing these enabling technologies, a diverse range of smart city applications can be developed. Numerous companies have already emerged in this domain, focusing on the cultivation and dissemination of innovative applications. For instance, the Smart Cities Connect Media and Research group is actively involved in promoting and advancing the field of smart cities.

On the other hand, laser technology has undergone significant advancements since its introduction in the 1960s. Due to its exceptional measurement accuracy, it finds extensive applications in various domains including military, industry, construction engineering, agriculture, and forestry. In the early 1980s, Nelson, Ross, et al. pioneered the use of airborne LiDAR to measure vertical features of forests, enabling the estimation of tree height and ground distance. Their results demonstrated that the tree height derived from laser data exhibited an error of less than 1 m compared to photogrammetry techniques [1]. Schreier employed airborne LiDAR to scan forested areas and successfully distinguished between the ground and ground vegetation based on the laser point cloud [2]. By leveraging distance information, reflectance, and other parameters, coniferous and broadleaf forests could be differentiated. As the 1990s approached, the detection technology of airborne LiDAR gradually matured. In recent years, the focus of research has shifted towards high-precision map model construction and target identification. Merlijn Simonse investigated the application of airborne LiDAR for environmental resource surveys in forestry [3]. Aleksey Golovinskiy conducted target identification in urban environments using a shape feature approach [4]. Their methodology involved obtaining 3D point cloud data in urban settings, performing point cloud segmentation using clustering methods to distinguish foreground and background entities, extracting shape features, and employing support vector machines for target recognition. Despite the existence of various methods for effectively recognizing laser point clouds in current research, several challenges, such as accuracy and robustness, still hinder the practical implementation of LiDAR in production settings.

Therefore, the development of 3D laser point cloud segmentation is crucial for enhancing risk resistance and improving management efficiency in smart cities. It is particularly necessary to utilize state-of-the-art mathematical models to effectively assist in enhancing the performance of intelligent systems. Numerous studies have made significant efforts in processing massive point cloud data and analyzing urban forests. Specifically, various intelligent algorithms have been developed to acquire professional feature knowledge and address specific challenges in city management applications. These algorithms employ diverse deep learning models, such as convolution, recurrent, attention, coder–decoder, and graph theory models. Subsequently, these interdisciplinary models are applied to make intelligent decisions related to various city tasks in designated garden and forestry scenes, including environmental factor prediction, visual classification, object detection, and segmentation. This significantly contributes to improving the connectivity, security, and effectiveness of smart city management.

Deep learning technologies have been extensively researched by scholars in the past decade and have yielded promising results compared to traditional methods in various urban management applications. However, accurately segmenting point clouds in scenes depicting urban environments remains a challenging task, influenced by numerous internal and external variables.

Volumetric-based approaches typically involve converting point clouds into 3D grids and applying 3D convolutional neural networks (CNNs) for shape classification. The earliest model for voxel identification, utilizing 3D CNNs, was proposed in the 2015 CNNS [5–7]. However, its progress has been hindered by the substantial computational requirements and the sparsity of stereo data after rasterization. FPNN [8] and Vote3D [9] proposed some solutions, yet difficulties persist when dealing with large volumes of point cloud data. Methods based on volume representation often produce coarse results, as only a small fraction of voxels contain information, making it nearly impossible to obtain detailed context within each voxel. Striking a balance between resolution and computation proves challenging in practical applications.

The multi-view method initially projects the 3D shape into multiple views and extracts view features. Subsequently, accurate shape classification is achieved by fusing these features. The primary challenge faced by these methods lies in the aggregation of multiple view features into a discriminative global representation. In the domain of point cloud deep learning, multi-view methods are extensively employed in shape classification research. Multi-view CNNs [10] aim to transform 3D point clouds or shapes into 2D images and utilize 2D convolutional networks for classification. This approach achieved state-of-the-art recognition results at that time owing to the mature development of 2D convolutional neural networks. However, this method encounters difficulties when extended to large scenes and 3D tasks such as point cloud classification. GVCNN [11] is capable of dividing a set of views into different groups based on their discriminative scores, which are obtained through quantized views. Prediction is then made by aggregating within groups and fusing between groups. Nevertheless, 2D multi-view images only serve as approximations of 3D scenes and fail to provide a true and lossless representation of 3D scenes, thereby resulting in a loss of geometric structure. Consequently, this can lead to less-than-ideal results in complex tasks such as point cloud semantic segmentation. Moreover, due to the intricate point information and surface details presented in point clouds, multi-view images cannot fully capture these pieces of information.

Pointwise MLP methods utilize multiple shared multilayer perceptrons (MLPs) to independently model each point and then aggregate global features using symmetric aggregation functions. Point-based deep learning methods have been extensively explored in recent years. In 2017, researchers at Stanford University introduced the PointNet architecture [12], which directly processes unordered point clouds as input data for recognition and semantic segmentation tasks. The network employs maximum pooling as a symmetric function to handle the disorder of point cloud models. To maintain the spatial invariance of the point cloud data, the structure incorporates two transformation matrices. However, PointNet is limited in its ability to capture local information of the model. As an improvement, PointNet++ was proposed [13]. By leveraging Farthest Point Sampling (FPS) and Multi-Scale Grouping (MSG), PointNet++ can extract local information and introduce PointWeb [14], a novel method for extracting contextual features from the local neighborhood of point clouds. PointWeb aims to establish tight connections between each point within the neighborhood, enabling more accurate representation of the region. To capture point interactions, an innovative feature-adaptive adjustment (AFA) module is introduced. Another approach, PointSIFT [15], draws inspiration from the SIFT algorithm to perform point convolution. Additionally, the Structure Relationship Network (SRN) was proposed by SRN [16] to enhance the performance of PointNet++.

Compared to convolutional kernels defined on two-dimensional grid structures, such as images, designing convolutional kernels for three-dimensional point clouds is challenging due to their irregularity. Current methods for three-dimensional convolution can be categorized into continuous convolution and discrete convolution, depending on the type of convolutional kernel used. PointCNN [17], a method based on point convolution, addresses the difficulty in applying convolution operations to irregular and disordered point cloud data by employing a technique called X-transformation. Flownet3D [18] follows the approach of feature extraction followed by feature correlation calculation. CNN methods

like the spherical convolution Spectral CNN [19,20] are currently limited to recognizing objects with organic-like and rich lattice-like structures, making it challenging to apply them to non-isometric shapes. KPCONV [21] proposes a novel point cloud convolution based on Kernel Points. It considers Kernel Points as reference objects for each point and calculates the weights using Geo-CNN [22], which introduces another method for modeling the geometric relationship between neighboring points. Geo-CNN achieves this by dividing the space into eight quadrants using six orthogonal bases. In PointConv [23] and MCCNN [24], the convolution operation is defined as a Monte Carlo estimation of hidden continuous three-dimensional convolution via importance sampling. Overall, the aforementioned methods address the challenges of designing convolutional kernels for three-dimensional point clouds and provide innovative solutions for effectively processing and analyzing such data.

Graph networks treat point clouds as vertices of a graph, where each vertex represents a point, and edges are generated based on the neighbors of each point. Features are learned in the spatial or spectral domain. The concept of treating each point as a vertex of the graph and generating connecting edges between neighboring points was first proposed by ECC [25]. DGCNN [26] utilizes MLPs to implement EdgeConv and symmetrically aggregates edge features on the neighborhood of each point, allowing for dynamic graph updates after each layer of the network. Several graph-based methods have been developed in this field. SuperPointGraph [8] employs gated neural networks to extract features. GCNN [27] and KPCONV [21] utilize graph convolution for effective processing of point clouds. ClusterNet [28] generates rotation-invariant features from each point and constructs a hierarchical structure of point clouds using an unsupervised approach. The features of subclusters at each level are learned through EdgeConv blocks and then aggregated through max pooling.

The current algorithms for processing point cloud data, particularly in terms of point cloud recognition, exhibit relatively low efficiency. These algorithms consume significant amounts of time during the batch extraction of parameters and require large memory usage. Consequently, effectively identifying point cloud data from large-scale outdoor scenes remains challenging. Thus, the development of new theories and algorithms is crucial in addressing this issue.

Regarding large-scale-scene point cloud data collected in garden environments, one encounters several challenges, including scene complexity, extensive computational requirements, uneven sparsity, as well as inherent characteristics such as disorder, rotation invariance, sparsity, severe occlusion, and an unstructured point cloud. Existing methods often struggle with limited scalability, excessive parameter complexity, and low segmentation accuracy in practical applications. To address these issues, we propose a novel point cloud deep learning framework called PointDMS. This framework consists of two main components: a multi-feature pre-processing segmentation module called DMS and a deep learning network named PointDMS. The PointDMS framework introduces the following innovations:

Features a multi-sensor acquisition system and data processing procedure to collect point cloud data in various terrestrial environments. This system facilitates the creation of a high-precision semantic annotation DMS dataset, specifically designed for deep learning training;

Introduces a novel DMS module, which is aimed at the rapid pre-processing and pre-segmentation of point clouds. The module effectively preserves the geometric features of complex and large-scale point cloud data;

Proposes a deep neural network, named PointDMS, which is trained to effectively segment and recognize point clouds that have undergone pre-processing using the DMS module.

These advancements contribute to the field of deep learning in point cloud analysis and pave the way for further research in this domain.

## 2. Materials and Methods

In this section, we sequentially present the labeling of the data acquisition system and DMS dataset, the module for pre-segmentation of multi-featured point clouds (DMS module), and the point cloud segmentation method, PointDMS, in the context of a garden scene.

### 2.1. DMS Dataset in Urban Forestry Scenes

This section provides an introduction to the relevant parameters of the laser scanner used for collecting point cloud data and its measurement principle. The point cloud data were obtained from static radar scanning and mobile radar scanning, as depicted in Figure 1. The upper section of Figure 1 illustrates vehicle-mounted laser radar scanning and backpack laser radar scanning, while station-mounted laser radar scanning is shown below. Static radar scanning is primarily utilized for capturing campus landscape scenes, resulting in dense point clouds. On the other hand, dynamic scanning is employed to gather information about the landscape of Beijing World Park, resulting in sparse data. The point cloud data were subjected to pre-processing and manual labeling using Geomagic software to generate manually classified semantic labels.

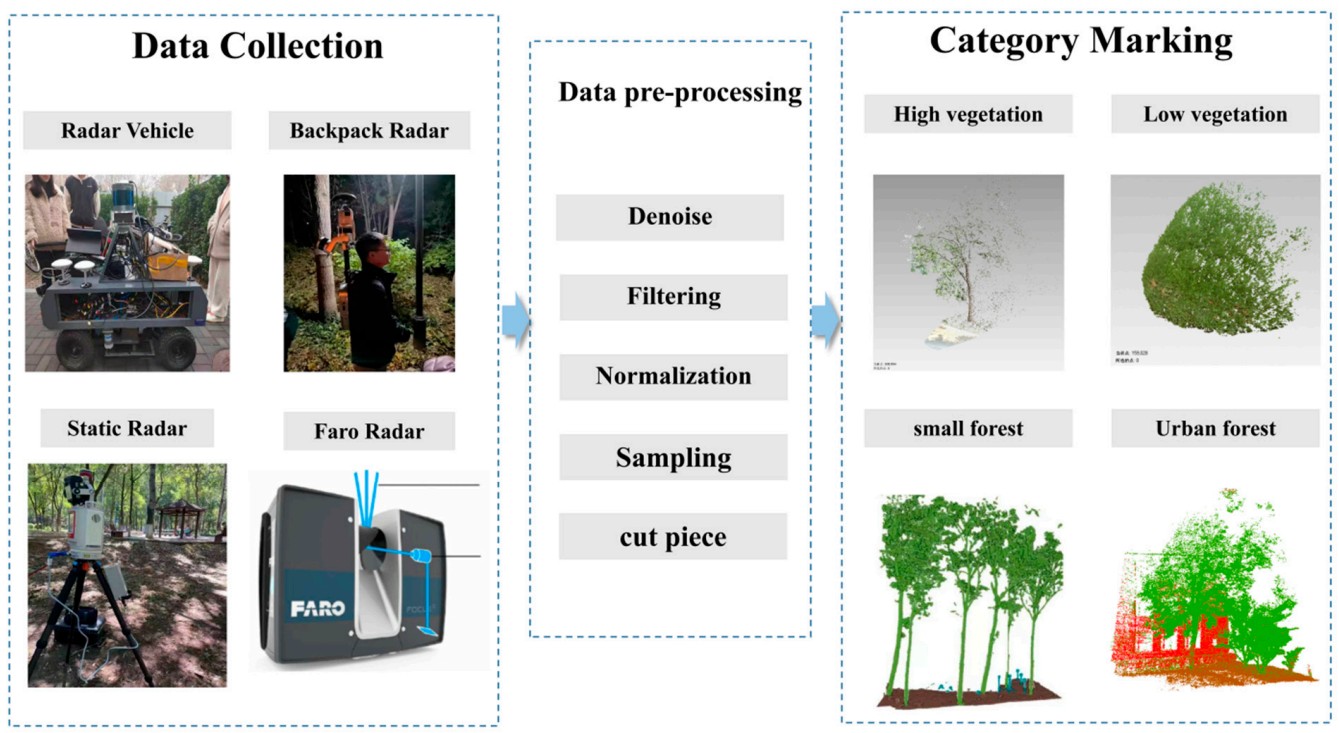

**Figure 1.** Point cloud data acquisition, pre-processing, and category marking for urban forestry scenes.

In the 1990s, similar to the GPS spatial positioning system, 3D laser scanning emerged as a groundbreaking technology in the field of surveying and mapping. This non-contact, high-speed laser scanning method enables the acquisition of geometric data, including high-resolution 3D coordinate data of large-scale object surfaces. It offers researchers a rapid means to reconstruct 3D models of objects. By utilizing a 3D laser scanner to gather information from sampled points, a point cloud can be generated.

The laser scanner operates on the principle of time of flight. A laser generator emits pulses of laser beams, which are directed towards a rotating optical mirror. This enables the laser beam to rotate in both vertical and horizontal directions. When the laser beam pulse encounters an object and reflects back, a receiver captures the reflected laser. By calculating the time of flight, the distance can be determined. Through the continuous rotation and emission of laser pulses, the scanner can obtain a wide range of distances surrounding it.

The data collected in this study were obtained using a variety of laser scanners. Firstly, we utilized our own multi-sensor-fusion LiDAR cart, which consisted of one 16-line LiDAR system, one 80-line LiDAR system, one GPS, one IMU, one camera, and other components. An embedded Nvidia Jeston host with ROS installed was employed to acquire multiple data streams, while SLAM was utilized to generate point cloud maps. This setup proved to be suitable for efficiently acquiring point cloud data in garden road environments. Secondly, we developed a backpack LiDAR system that included two 16-line LiDAR systems for collecting horizontal and vertical point clouds, respectively, an integrated IMU, a battery, and an embedded Linux device. By employing ROS for multi-sensor data acquisition and SLAM for point cloud mapping, this system was well-suited for collecting forest information in garden environments and capturing the mobility characteristics of pedestrians. Additionally, we utilized RIEGL rack-stand scanning (3) and FARO scanning as shown in Table 1. (4) as fixed-point scanning methods. These techniques allowed us to obtain high-precision point cloud data with detailed information as well as large volumes of data. The equipment used in this study is depicted in Figure 1, and the data collection process is illustrated in Figure 2.

**Table 1.** FARO focus3D s350 Scanner equipment specifications and parameters.

| Instrument Modules | Indicators | Specific Parameters |
|---|---|---|
| Laser | Laser Class<br>Wavelength<br>Beam divergence<br>Beam diameter (exit) | Level 1<br>1550 nm<br>0.3 mrad (0.024°)<br>2.12 mm |
| Beam diameter (exit) | Accuracy<br>Angular accuracy<br>Range<br>Measuring speed (pts/sec)<br>Ranging Error | 1 mm<br>Horizontal + Vertical:19 arcs<br>0.6–350 m, Indoor or outdoor<br>122,000/244,000/488,000/976,000<br>$\pm$1 mm |
| Color unit | Resolution<br>HDR<br>Parallax | Maximum 165 megapixel color<br>$2\times, 3\times, 5\times$<br>Coaxial design |
| Deflection unit | Field of view (vertical/horizontal)<br>Step size | 300°/360°<br>0.009° (40960 3D-Pixel on 360°)<br>5820 rpm 97 Hz |
| Multi-sensor | Dual-Axis Compensator<br>Altitude Sensor<br>Altitude<br>Compass<br>GPS | Leveling each scan: Accuracy 0.019°; size $\pm$ 2°<br>Detection of the height relative<br><2000 m<br>Electronic compass provides<br>Integrated GPS receiver |

We employed uniform downsampling. Uniform downsampling involves a sphere with a radius of r utilized for uniform sampling. The sampling point closest to the center of the sphere is selected to replace the points within the sphere. The point cloud space is divided by sampling every fixed number of points, following the order of the points. Point cloud denoising is a crucial pre-processing step aimed at eliminating noise and outliers from 3D point cloud data. Due to inherent limitations of 3D scanning devices and imperfect image reconstruction techniques, point cloud data are often prone to noise interference. The objective of denoising methods is to restore the true structure of point clouds while preserving their key features. This typically involves analyzing each point in a point cloud to determine if it could be noise and adjusting or removing it based on the attributes of its neighboring points. Efficient point cloud denoising not only enhances the accuracy of subsequent tasks such as 3D reconstruction, classification, and recognition but

also provides clearer and more accurate data for advanced 3D analysis and applications. Denoising algorithms are based on local characteristics of point clouds, such as point density, neighborhood relationships, and surface curvature. Common methods include the following: statistics-based denoising, where the spatial distribution of neighbors for each point is analyzed to determine if it is noise; filtering-based methods, such as Gaussian filtering or median filtering, used to smooth point clouds and reduce random noise; and model-based methods, where surface fitting or other geometric models are employed to estimate the true point cloud structure and eliminate points that deviate from the model. The objective of these algorithms is to eliminate noise while preserving the original structure and details of point clouds. We identified outlier points based on the distance distribution between each point and its neighbors. By calculating the average distance and standard deviation and defining a threshold based on these statistical measures the algorithm could effectively identify and remove noise and outlier points. This approach is a commonly used method in point cloud denoising due to its simplicity, intuitiveness, and effectiveness. Subsequently, the obtained point cloud data were segmented and filtered using Geomagic software [29]. Directly processing the large volume of data collected by static LiDAR is challenging. We segmented the point cloud into multiple smaller scenes and then sampled the segmented point cloud. Finally, all groups of individual target data were saved as six-dimensional color point clouds, three-dimensional point clouds, and corresponding label data with MATLAB. Consequently, based on the measured forest environment data, a total of 30,215 sets of data were extracted from 5 categories, including (1) trees, (2) low shrubs, (3) buildings, (4) roads, and (5) unclassified points, constituting a multi-dimensional laser point cloud database of urban forest scenes. This process is visualized in Figure 1.

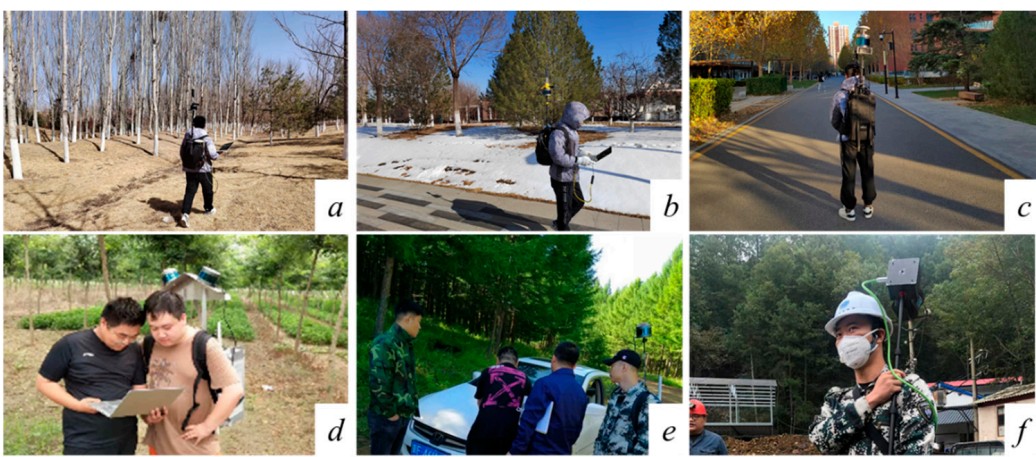

**Figure 2.** Point cloud acquisition equipment and data collection processing at different sites: (*a*) Gaotang County Maubai Poplar Base, (*b*) Beijing World Park, (*c*) Beijing Forestry University, (*d*) Gaoyang County Forestry Base, (*e*) Sehanba State-owned Mechanical Forestry Field, and (*f*) Luoyang Luanchuan City State-owned Forestry Area).

The point cloud data utilized in this study were measured by the Special Equipment Research Center of Beijing Forestry University, Using the laser radar in Table 2 for data collection. Data collection and dataset establishment were conducted at six different sites, as shown in Figure 2. Despite the initial sampling process, there still remained over 1299 million points in the complete point cloud datasets. In order to construct a database suitable for deep neural networks, independent datasets of target objects, obstacles, weeds, etc., needed to be created. Therefore, further processing, such as segmentation, was required for the simplified data, followed by labeling and database creation. The objects were categorized into five categories, namely trees, low shrubs, buildings, roads, and unclassified points.

**Table 2.** Line scanner equipment specifications and parameters.

| Training Hyperparameters | Parameter Values |
|---|---|
| Sensors | TOF method ranging 16 channels |
| | Measurement: 40 cm to 150 m (20% target reflectivity) |
| | Accuracy: ±2 cm |
| | Angle of view (vertical): ±15° (30° total) |
| | Angular resolution (vertical): 2° Viewing angle (horizontal): 360° |
| | Angular resolution (horizontal/azimuth): 0.1° (5 Hz) to 0.4° (20 Hz) |
| | Speed: 300/600/1200 rpm (5/10/20 Hz) |
| Laser | Class 1 |
| | Wavelength: 905 nm |
| | Laser Emission Angle (full angle): 7.4 mrad horizontal, 1.4 mrad vertical |
| | ~300 k dots/s |
| Output | 100 Gigabit Ethernet |
| | UDP packets contain |
| | Distance information |
| | 16 line parameters |
| | Rotation angle information |
| | Calibrated reflectivity information |
| Mechanical/electronic operation | Power consumption: 12 w (typical) |
| | Operating voltage: 9–32 VDC (requires interface box and stable power supply) |
| | Weight: 0.87 kg (excluding data cable) |
| | Operating voltage: 9–32 VDC (requires interface box and stable power supply) |
| | Weight: 0.87 kg (excluding data cable) |
| | Dimensions: Diameter 109 mm × Height 80.7 mm |
| | Protection and safety level: IP67 |
| | Operating ambient temperature range: −30 °C~60 °C |
| | Storage ambient temperature range: −40 °C~85 °C |

　　　To address the issue of the small proportion of forest point clouds in garden environments, we implemented a data expansion approach. This involved expanding the existing tags, such as the green tags and blue tags shown in Figure 3, based on data volume. For the important identified forestry information, we expanded the volume by a factor of 10 for trees and by a factor of 5 for shrubs. Additionally, a 50-fold rotation of the point clouds was performed for data expansion. By considering features such as point cloud scattering and other parts that describe crown-type features more prominently, we achieved a 15-fold expansion to obtain the training data.

　　　Semantic3D [30] is a dataset specifically designed for large-scale outdoor environments. Each frame in the dataset represents a single frame of data obtained from a fixed position using a ground-based LiDAR scanner. The dataset primarily consists of point clouds categorized into classes such as ground, vegetation, and buildings. Each point in the dataset contains RGB and intensity information. The dataset covers both rural and urban scenes and classifies categories such as ground, vegetation, and buildings into 8 semantic classes. As a benchmark dataset for large-scale point cloud classification, the tDt cloud dataset includes over 4 billion points and encompasses a variety of urban scenes. We utilized this dataset as a reference to construct our own DMS dataset for relevant research. Figure 3 showcases samples from our collected DMS dataset, which includes 1150 small scenes from the campus of Beijing Forestry University and Beijing World Park, encompassing a total of 12,530 environmental scenes. Our DMS dataset comprises a total of 13,680 scenes. Specific details are presented in Table 3, where a comparison with other point cloud datasets is provided. Our dataset focuses on urban park environments and encompasses a wide range of points, including a greater variety and quantity of live trees. We employed uniform tags:

green represents trees, brown represents ground, red represents buildings, blue represents low shrubs, and so on. This dataset can be effectively utilized in deep learning research focused on point clouds in garden environments. Parts of the DMS dataset have been made available on the Baidu cloud disk. Interested individuals can download it for research purposes using the following link: https://pan.baidu.com/s/1RxQVL89aqBJ_gdioXU52Mw, access on 17 October 2023, Extraction code: 5554.

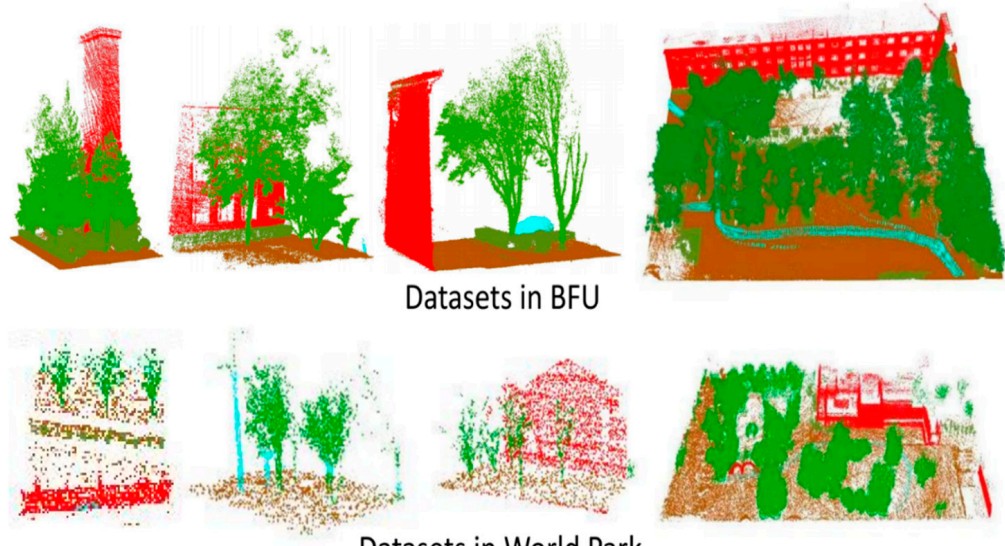

**Figure 3.** The campus scene of Beijing Forestry University and the Beijing World Park scene in the DMS dataset, where green represents trees, brown represents ground, red represents buildings, blue represents low shrubs.

**Table 3.** Comparison of different point cloud datasets.

| Dataset | Scenes | Points (Millions) | Classes | Sensor | Annotation |
|---|---|---|---|---|---|
| SemanticKITTI [31] | 23,201 | 4549 | 25 | Velodyne HDL-64E | point-wise |
| Oakland3d [32] | 17 | 1.6 | 5 | SICK LMS | point-wise |
| Freiburg [33] | 77 | 1.1 | 4 | SICK LMS | point-wise |
| Wachtberg [34] | 5 | 0.4 | 5 | Velodyne HDL-64E | point-wise |
| Semantic3d [30] | 15/15 | 4009 | 8 | Terrestrial Laser Scanner | point-wise |
| Paris-Lille-3D [35] | 3 | 143 | 9 | Velodyne HDL-32E | point-wise |
| KITTI [36] | 7481 | 1799 | 3 | Velodyne HDL-64E | bounding box |
| DMS dataset | 13,680 | 1299 | 5 | RS16E; Faro64E | point-wise |

### 2.2. Pre-Segmentation Module DMS

Regarding urban forestry scenes, the dataset contains a large number of points, which poses a significant computational burden. Moreover, the complexity of garden scenes' cloud environments and the uneven sparsity of point clouds across different areas make it challenging to directly apply deep learning methods for computation. To address these issues, we implemented a semantic pre-segmentation module called the DMS module specifically designed for garden scenes. This module was chosen to tackle the problems of

excessive input and uneven distribution of semantic information. Please refer to Figure 4 for a diagram illustrating the proposed module.

$$\text{Linearity} = \frac{\lambda_1 - \lambda_2}{\lambda_1} \tag{1}$$

$$\text{Planarity} = \frac{\lambda_2 - \lambda_3}{\lambda_1} \tag{2}$$

$$\text{Scattering} = \frac{\lambda_3}{\lambda_1} \tag{3}$$

We utilized the geometric feature descriptors of point clouds [37], computed as Equations (1)–(3). Among these descriptors, linearity represents the linear geometric feature of a point cloud, indicating the extent of linear stretching of the point cloud neighborhood. Flatness evaluates the point cloud's fit to a plane, while scattering corresponds to the spherical neighborhood, describing its isotropic features. These features serve as dimensional characteristics of the point cloud. To generate redundant point cloud features, we introduced an attentional aggregation module for point cloud features called the DMS module.

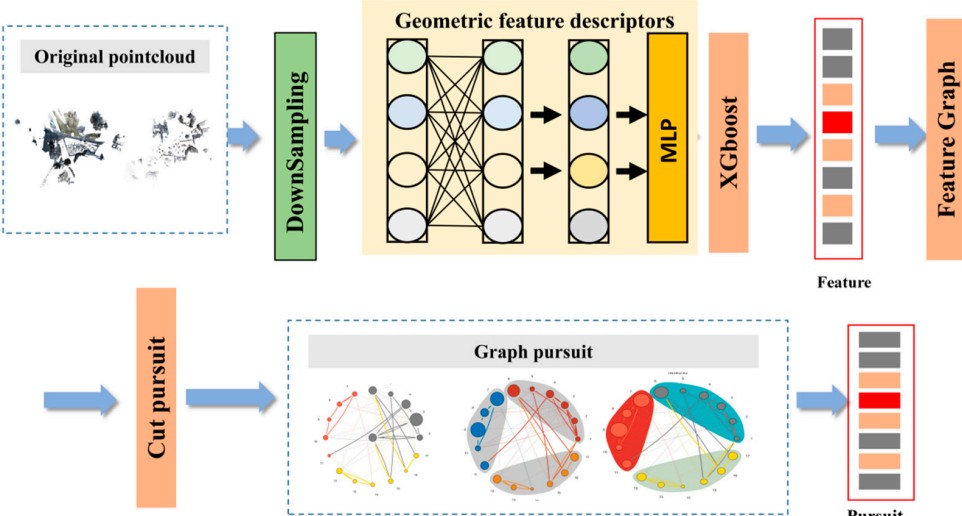

**Figure 4.** Diagram of the DMS module. This module is used for feature aggregation and efficiently merges point cloud features and oversamples them into smaller blocks with similar characteristics.

To construct redundant point cloud features through k-machine downsampling and incorporate the geometric feature descriptors of point clouds into subsequent deep neural networks, we utilized an attention-based aggregation module for point cloud features: the DMS module. The module employed the k-nearest neighbors algorithm to find the k nearest neighboring points for each point in Euclidean space and calculated the geometric feature descriptors. V represents the vertical characteristic descriptor of the point cloud, describing its vertical plane characteristics [37]. Centered on each neighborhood geometric feature descriptor and point, according to Equations (4)–(6), this feature is considered redundant and can effectively minimize losses. Geometric feature descriptors establish connections between the center point of each neighborhood and the geometric features of that point through shared weights in MLP, resulting in the creation of a new point feature.

$$f_i^k = MLP\left( f_{i=1}^{i=k} L \oplus f_{i=1}^{i=k} s \oplus f_{i=1}^{i=k} P \oplus f_{i=1}^{i=k} V \right) \tag{4}$$

$$\hat{f}_i^k = concat\left( f_i^k, f_i L, f_i s, f_i P, f_i V \right) \tag{5}$$

$$score_i^k = a\left(\hat{f}_i^k, W\right) \tag{6}$$

$$\widetilde{f}_i = \sum_{k=1}^{K}\left(\hat{f}_i^k \times score_i^k\right) \tag{7}$$

We employed the max pooling technique to automatically learn the selection of valuable information from the combined cascaded features using the attention method. The XGBoost [38] feature filter was then utilized to extract the relevant features, resulting in the acquisition of new aggregated features. By integrating the aforementioned features, we constructed an unsupervised graph of the output features within the DMS module, aiming to establish an over-segmented unsupervised model.

In the XGBoost model, we chose the gbtree model for iteration and set a higher initial learning rate of 0.1. The ideal number of decision trees was determined to be 50. To optimize the decision tree-specific parameters and the regularization parameter of XGBoost, we gradually decreased the learning rate and identified the optimal parameters. Eventually, we obtained the parameters subsample = 0.6 and colsample_bytree = 0.8.

To enhance the feature representation, we introduced a function to learn an additional score for each feature. Additionally, by sharing the learnable parameters of the MLP, an additional independent score was learned for each data point. By leveraging this software mask for automatic feature selection, we were able to obtain a weighted sum at the level of neighborhood feature points, as depicted in Equation (7).

$$Q_{(x)} = \underset{g \in \mathbb{R}^{4 \times V}}{\arg\min} \sum_{i \in V} \|g_i - f_i\|^2 + \rho \sum_{(i,j) \in E} \delta\left(g_i - g_j \neq 0\right)^{V \times 4} \tag{8}$$

In order to characterize each point, we utilized its local geometric feature vector, which was derived from the aggregated features mentioned earlier. Our objective was to solve the optimization problem [20] by optimizing the solution of $Q_{(x)}$ (Equation (8)). To address the problem, we employed the concept of greedy reduction [12] on the 3D point cloud. The following procedure outlines the energy-optimized solution for the integrated 3D point cloud aggregation feature.

$$G = (V, E, \omega) \tag{9}$$

Equation (9) defines an undirected weighted graph, where E represents the set of edges and p represents the weight. In this context,

$$Q(x) = f(x) + \frac{\lambda}{2} \sum_{(i,j) \in E} w_{ij}\phi(x_i - x_j) \tag{10}$$

where the function $f(x)$ is a differentiable function, $\phi$ is a penalty function, and the problem we need to solve is the problem of minimization of $Q_{(x)}$. Landrieu, in 2017, proposed a strategy to minimize differentiable functions that construct a full-variational half-table regularization on weighted graphs. In this study, we utilized an improved scheme based on the spg [39] method by expanding the algorithm's scope to include functions with non-differentiable parts separated along the graph vertices. Our objective was to identify smooth points of the function F, where the differential was zero but the left and right derivatives were positive or negative. These smooth points represented points where all directional derivatives were non-negative. It is worth noting that the aforementioned assumption held true if all considered generalized functions were convex, and a smooth point was considered equivalent to a global minimum value.

$$S_i(x) = \left\{(i,j) \in E \,\middle|\, x_i \neq x_j\right\} \tag{11}$$

Taking edges supporting the above gradient as a set S, S collects the characteristics of the above edges.

$$dF(z) = \sum_{v \in V} \delta_v(z)d_v + \sqrt{\sum_{(i,j) \in E} \rho_{ij}\phi|d_i - d_j|^2} \tag{12}$$

We wanted to consider the problem of minimization of a convex differentiable function. We implemented a working set algorithm to solve the above problem of minimizing the $Q_{(x)}$ function.

$$d_V(z) = \tilde{N}f(z) + \sum_{\substack{(e,u)\hat{I}E \times V \\ e = (u,v)or(v,u)}} r_e sign(z_v - z_u) \tag{13}$$

The cut tracking algorithm iteratively optimized the partition $V$ initialized at $\{V\}$. In each iteration, the approximate problem corresponding to the current partition was solved and its solution was used in turn to refine the components of $V$.

$$Q'(z, 1_B) = \langle \nabla Q_C(z), 1_B \rangle + \lambda \omega s^{\{E-C\}}(B, B^{\{E-C\}}) \tag{14}$$

It can be shown that the directional derivative in the direction of $B$ is defined as

$$Q'(z, u_B) = (\gamma_B + \gamma_{B\{E-C\}})Q'(z, 1_B) \tag{15}$$

*2.3. PointDMS Framework*

Based on our proposed PointDMS fusion module, we introduce a novel semantic segmentation network in this study. The network's framework is illustrated in Figure 5, representing an end-to-end input framework. At the input end, we have the original point cloud, the real labels of the original point cloud, and the pre-segmented point cloud clusters formed by DMS. These clusters integrate the local information of the point cloud and are input into the neural network for computation. The left side of the figure displays the input point cloud, where PointDMS includes the original point cloud and the point cloud segmented by the DMS module. In the middle, different neural network layers process the original point cloud and the pre-segmented point cloud regions. The network structure for processing remains the same. On the right side of the network, the output consists of the point cloud labels predicted by the network. On the input side, we provide finely semantically labeled point clouds, establish a greedy reduction function, and obtain an over-segmentation module. This module contains the line segmentation graph of the main features of the point clouds. The features are then passed to the T-Net for rotation. T-Net, a $64 \times 64$ matrix proposed in PointNet, multiplies the input point cloud, and after a round of convolutions and fully connected layers, a matrix representing the features of the data is obtained. This matrix, when multiplied with the previous input, ensures transformation invariance and preserves the transformation invariance of the point cloud to some extent. For the 10-million-scale point cloud, we first performed point cloud voxelization and then used a voxel feature encoder. The voxel feature encoder consisted of only a multilayer perceptron (MLP) and a maximum pooling layer, generating sparse voxel features.

$$f_i = \text{MaxPooling}_{f_i = vn}(\text{MLP}(p_1 \dots p_n)), i \in (1 \dots n) \tag{16}$$

$$N_i = a \cdot \text{Pointnet}_{n_i = n}(\text{MLP}(p_1 \dots p_n)), i \in (1 \dots n) \tag{17}$$

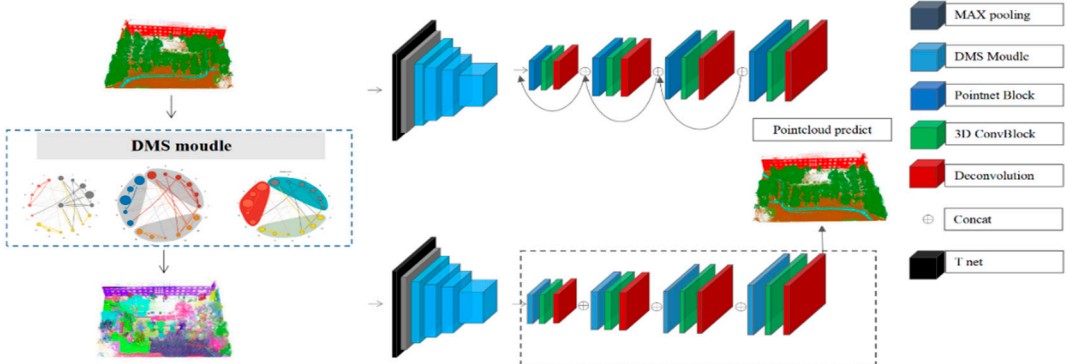

**Figure 5.** Structure of PointDMS, which has 3 main parts. In the left part, PointDMS receives the point cloud's multi-feature information aggregated by DMS module. In the middle part is the point cloud's over-segmentation based on energy entropy as well as the proposal of the type of point cloud according to the over-segmentation results. After going through PointNet and the fully connected layer, the point cloud is segmented into multiple classes, comprising the right part.

The original $N_i$ obtained by the multilayer point cloud feature extraction module of PointDMS represents different geometric partitions; each geometric partition was feature-averaged and extracted with PointNet for the global feature.

$$vi = k \cdot \text{Pointnet}_{n_i=n}(\text{MLP}(S_1 \dots S_n), \text{MLP}(E_1 \dots E_n)), i \in (1 \dots n) \tag{18}$$

For the segmented view of the point cloud, in Equation (18), $S_n$ and $E_n$ represent the point features and edge features, respectively, which constitute the contour features of the point cloud and better represent the overall structural features of the point cloud. Similarly, we constructed PointNet as a feature extractor for overall feature extraction.

$$Loss = \sum_{i \in \{fi, Ni, vi\}} \frac{1}{2\sigma_i^2} Loss_i + \frac{1}{2} \log \sigma_i^2 \tag{19}$$

This is a classic iterative optimization problem in the field of computer science. In this problem, the superpoint set and feature vectors correspond to the nodes and labels of a graph, respectively. As shown in Algorithm 1, the method described follows an alternating optimization strategy, where one set of variables is optimized while keeping the other set fixed, until the algorithm converges. Here is a brief overview of the algorithm's process: Initialization: assign an initial label to each node (superpoint) of the graph G. Iterative optimization: continuously iterate the following two steps until convergence. Optimize feature vectors: under the current label assignment, optimize the feature vectors of each superpoint s to minimize the loss function related to the total variation of the graph. Optimize labels: under the current feature vectors, update the labels of each node of the graph G to minimize the loss function related to the total variation of the graph. This involves using graph-cut algorithms (maximum flow/minimum cut) to find the optimal label assignment. Semantic label assignment: assign a semantic label to each point in the original point cloud P based on the final determined labels and feature vectors. Output: obtain the semantic segmentation result of point cloud P. As shown in Figure 5, our algorithm is an end-to-end processing framework. Firstly, the original point cloud is inputted, and T net is used to generate different geometric blocks. The point clouds in different regions are then batched into the neural network, and PointNet is used as the feature extractor to obtain the local comprehensive features of each over-segmentation module and the original features vi before over-segmentation. These two features are part of the auxiliary segmentation. Among them, Ni represents the local geometric features of the point cloud, and their geometric features have the best energy entropy similarity in each local geometric region. Vi represents the overall geometric features. Firstly, we convert the above point cloud into

h5 format for easy computer processing. Secondly, we divide it into blocks of size 5000. Thirdly, we perform rotation pre-processing and point cloud feature rotation and put them into the DMS module for feature extraction. Finally, we perform feature extraction through multiple layers of PointNet. Three-dimensional convolution layers are used for feature integration, and semantic segmentation is performed using the interpolation method as shown in the figure. Finally, a multi-task loss function is constructed, as shown in Equation (19), which includes segmentation loss, over-segmentation block loss, and neighborhood segmentation graph loss. The coefficients of the semantic segmentation loss are equal.

---

**Algorithm 1:** Greedy graph-cut pursuit algorithm to optimize 3D point cloud over-segmentation.

---

Extraction of point cloud features linearity, planarity, and scattering

Aggregation of effective point cloud features $\widetilde{f}_i$

Construct the energy function

$Q(\mathrm{x}) = \underset{g \in \mathbb{R}^{4 \times V}}{\text{argmin}} \sum_{i \in V} \|g_i - f_i\|^2 + \rho \sum_{(i,j) \in E} \delta\left(g_i - g_j \neq 0\right)^{V \times 4}$

Optimizing Q(x), create node diagram

$G = (V, E, \omega)$

$S(x) = S \ S := S \cup (N \times N^c) \ span(\{\mathbf{1}_C | C \in V_{new}\})$

$V_{new} := \{L | \exists A \in V\}$ L is a connected comp. of $(A \cap N) \cup (A \cap N^c)$

$x_v \in \text{argmin} Q(z)$

Initialize $V \leftarrow \{V\}$

repeat

Pick $B \in \text{argmin}_{B' \subset V} Q'(x_V, \mathbf{1}_B)$

$V \leftarrow \{N \cap A\}_{A \in V} \cup \{N^c \cap A\}_{A \in V}$

$V \leftarrow$ connected components of elements of $V$

Pick $x_v \in \text{argmin}_{z \in span(V)} Q(z)$

find $\xi^V \in R^V$, stationary point of $F^V : \xi \to F(\sum_{U \in V} \xi_U 1_U)$

$x_v \leftarrow \sum_{U \in V} \xi^{(V)}_U 1_U$

find $d^{(x)} \in D$ minimizing $d \to F'(x, d)$

$V \leftarrow U_{U \in v}\{$ maximal constant with $(d_u^{(x)})_{u \in U}\}$

until $\min_{B \subset V} Q'(x_v, \mathbf{1}_B) < 0 \&\& F'(x, d^{(x)}) \geq 0$

return $(V, x_v)$

Input the optimized segmentation results into PointDMS training

$Loss = \sum_{i \in \{fi, Ni, vi\}} \frac{1}{2\sigma_i^2} Loss_i + \frac{1}{2} \log \sigma_i^2$

Obtain optimal segmentation results

---

## 3. Results and Discussion

### 3.1. Model Sets and Evaluation

In our DMS dataset, algorithm performance evaluation is conducted using the accuracy, IoU, and mIoU metrics, which are derived from the semantic3D [30] dataset. TP represents true positives, which are point clouds correctly predicted as belonging to a certain class. TN represents true negatives, which are point clouds correctly predicted as not belonging to a certain class. FP represents false positives, which are point clouds wrongly predicted as belonging to a certain class. FN represents false negatives, which are point clouds wrongly predicted as not belonging to a certain class. Accuracy is the ratio of correctly classified point clouds to the total number of point clouds, as shown in Equation (20). However, in the semantic segmentation of point clouds, accuracy may not be the most suitable metric due to potential class imbalance in point clouds. IoU is the ratio of the intersection area between the predicted segmented point cloud and the ground-truth segmented point cloud to their union area. Here, TP represents the common region between the predicted and ground-truth values of class C, while FP and FN represent the incorrectly predicted region

and the missed region, respectively, as shown in Equation (20). mIoU is the average IoU of all classes, and if there are N classes, mIoU is calculated as shown in Equation (22).

$$Accuracy = \frac{TP + TN}{TP + TN + FP + FN} \tag{20}$$

$$IoU(C) = \frac{TP}{TP + FP + FN} \tag{21}$$

$$mIoU = \frac{1}{N}\sum_{i=1}^{N} IoU(C_i) \tag{22}$$

The training accuracy and loss values of the model obtained from each iteration of the training process are depicted in Figure 6. As the number of iterations increases, the model's training accuracy gradually improves, while the training loss value gradually decreases. In the initial stage of model training, the model experiences higher learning rates, resulting in a faster convergence of the training loss curve. However, as the number of iterations increases, the slope of the training loss curve gradually diminishes. Eventually, the model training terminates. After 450 iterations, the training accuracy and loss curves stabilize, with the training accuracy reaching approximately 0.93 and the loss function reaching approximately 0.14. This indicates that the PointDMS model does not suffer from overfitting or underfitting gradient disappearance issues.

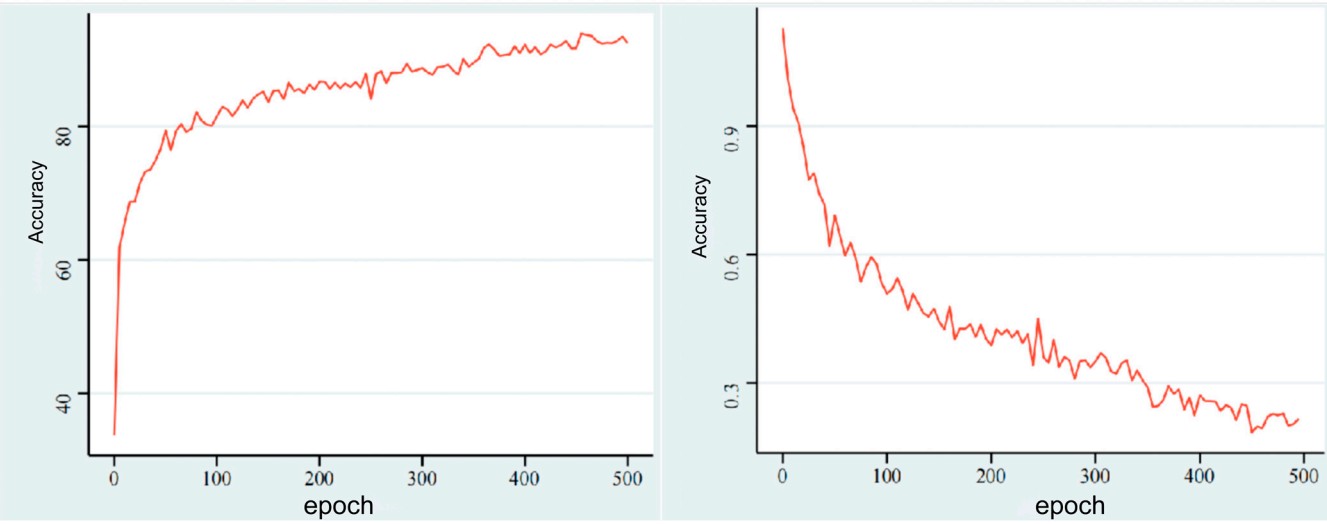

**Figure 6.** Training accuracy curve and loss curve of PointDMS.

### 3.2. PointDMS Data Processing Results

According to Algorithm 1, after applying the energy entropy optimization method for optimal point cloud segmentation, we obtain optimal point cloud over-segmentation chunks. These chunks ensure that the objects do not overlap semantically, while the boundaries of the over-segmentation chunks may overlap with the semantic boundaries. This approach effectively retrieves both relatively large environmental objects and relatively small grain segments.

In Figure 7, which represents the hyper-segmentation map of the campus garden scene "c" in the DMS dataset, we observe that the purple building is hyper-segmented into a single block, while the different trees and the ground are hyper-segmented into numerous small detailed blocks. This hyper-segmentation allows for the distinction between tall trees and low shrubs, as well as the separation of different objects into distinct parts based on their size.

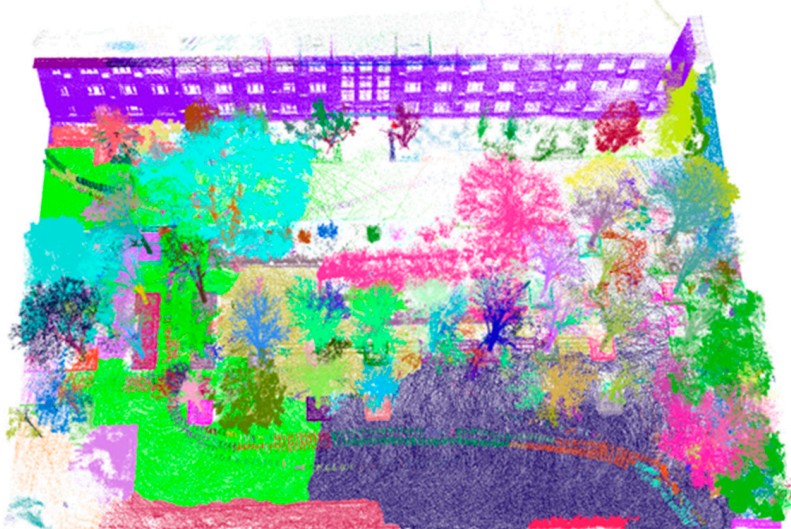

**Figure 7.** DMS dataset hyper-segmentation effect. Different adjacent colors represent different semantic blocks, and blocks with similar semantic information are allocated to the same block. For example, purple represents buildings with high semantic consistency, while individual trees are divided into multiple areas.

The hyper-segmentation effect of a portion of the urban forestry scene in the DMS dataset is evident in the separation of live trees from the ground (Figure 8). Various standing trees are fragmented into distinct small blocks, while still maintaining semantic consistency.

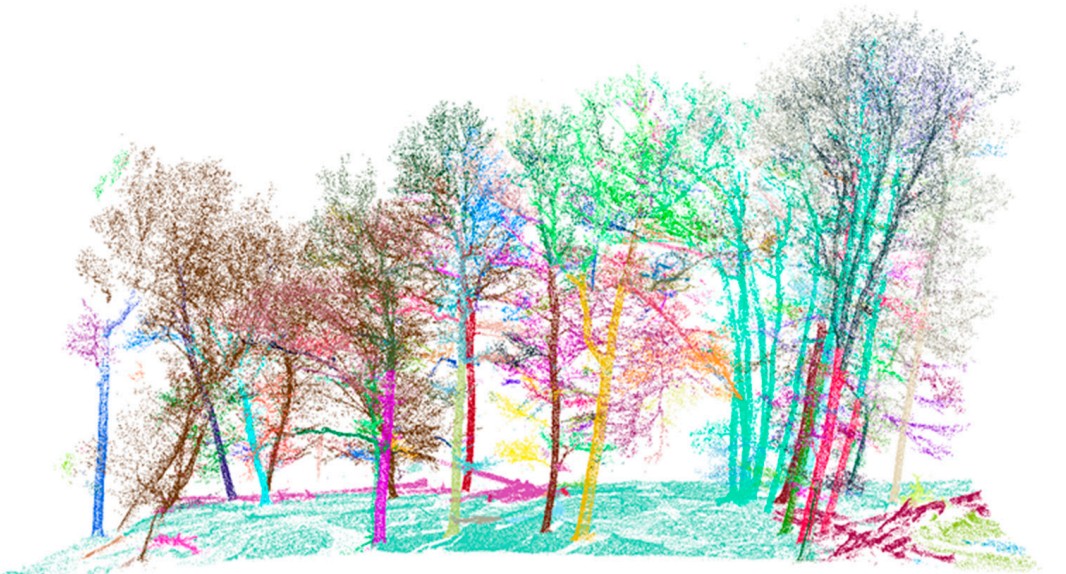

**Figure 8.** DMS dataset live tree hyper-segmentation effect.

Our segmentation of scene c in the DMS dataset, which represents a campus garden environment of Beijing Forestry University, is illustrated in Figure 9. It is evident that the buildings, as geometrically homogeneous objects, consistently maintain a certain level of integrity across the four smaller scenes. Additionally, the over-segmentation effect on trees and other sparse elements is noticeable. Objects with distinct geometric characteristics are categorized into different classes.

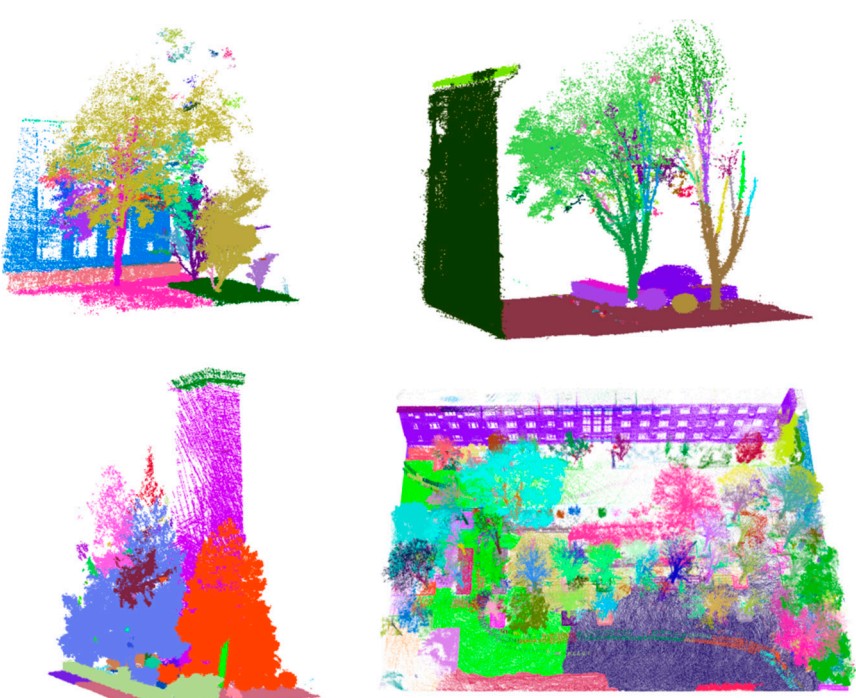

**Figure 9.** Example of PointDMS over-segmentation visualization of DMS dataset.

Our over-segmentation of scene b in the DMS dataset, which represents a garden environment of Beijing World Park, is depicted in Figure 10. In comparison to scene c, the point cloud data we gathered are sparser. It is evident that in the four small scenes, the impact of over-segmentation on sparse elements such as trees and shrubs is pronounced. Objects with distinct geometric characteristics are categorized into separate classes.

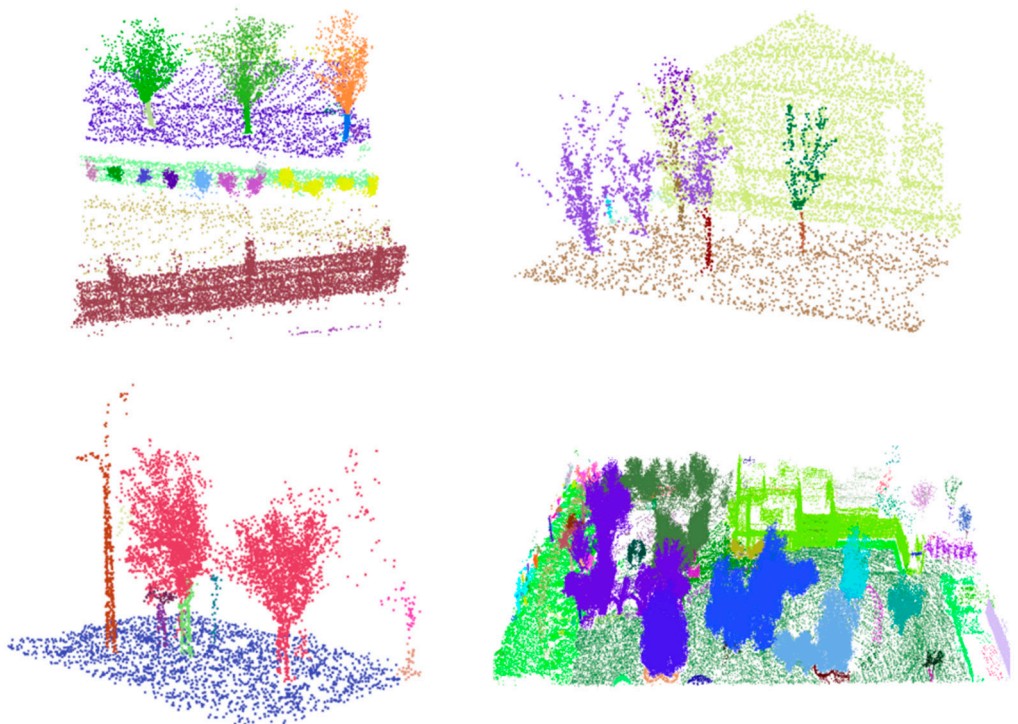

**Figure 10.** Example of PointDMS over-segmentation visualization of DMS dataset's scene b.

The hyperparameters presented in Table 4 were chosen for experimentation. The development environment consisted of Python 3.8 and PyTorch 1.3 installed on Ubuntu 18.04, utilizing a GPU with dual NVidia 1080ti graphics cards for CUDA-accelerated computation. Prior to training, the point cloud was randomly rotated along the Z axis and subjected to a 30% random dropout method. Random discarding of data during each training epoch effectively enhances the generalization of the training process, enabling the algorithm to perform well on sparse point clouds. For optimization, we employed the Adam optimizer due to its simplicity of implementation, computational efficiency, low memory requirements, and suitability for large-scale data and parameter scenarios. Additionally, we incorporated the optimal learning rate method during training, adjusting the learning rate based on the loss value. The hyperparameters utilized in our training process are outlined in Table 4.

**Table 4.** Parameter settings.

| Training Hyperparameters | Parameter Values |
| --- | --- |
| Initial KNN parameter selection | 10 |
| Maximum number of iterative steps | 5000 |
| Learning rate | 0.001 |
| beta1 | 0.9 |
| beta2 | 0.999 |
| Block Size | 80 |

*3.3. Large-Scale Point Cloud Segmentation Results*

The segmentation results of scene c in the DMS dataset are presented in Figure 11. This dataset consists of a point cloud capturing a campus garden environment, which was scanned multiple times at Beijing Forestry University. The upper section of the figure represents the true dataset labels, while the lower section displays the corresponding prediction results. Visually, the predicted point clouds closely resemble the labeled data. Notably, the segmentation of tree and shrub point clouds, distinguishing them from the overall scene, is accurate. However, there are some misclassifications in certain areas. For instance, in the first small scene, shrubs located near the ground under the wall are mistakenly identified as buildings. Additionally, in the second scene, a red building exhibits some sparse building point clouds that are incorrectly classified as trees. In the third scene, a small portion of the tree's top point cloud is misidentified as a building. Lastly, in the fourth scene, the lane is misclassified. These misidentifications primarily occur at the edges of the sample and in sparsely populated point clouds.

Segmentation results of the DMS dataset's scene b are depicted in Figure 12, showcasing point clouds of the World Park environment obtained from multiple scans. The upper portion represents the true dataset labels, while the lower portion displays the corresponding prediction results. These point cloud predictions exhibit a remarkable visual similarity to the labels, particularly in the accurate segmentation of tree and shrub point clouds from the overall point cloud. However, upon closer inspection, certain issues arise. In the first small scene, the upper left corner is erroneously identified as a building, and in the second scene, certain parts of tree trunks are misclassified. Furthermore, the third scene misidentifies the ground edge as a building, and in the fourth scene, a small section of trees in the right corner is falsely labeled as a building. It is worth noting that these misclassifications predominantly occur at the boundaries of different samples and within sparsely populated point clouds.

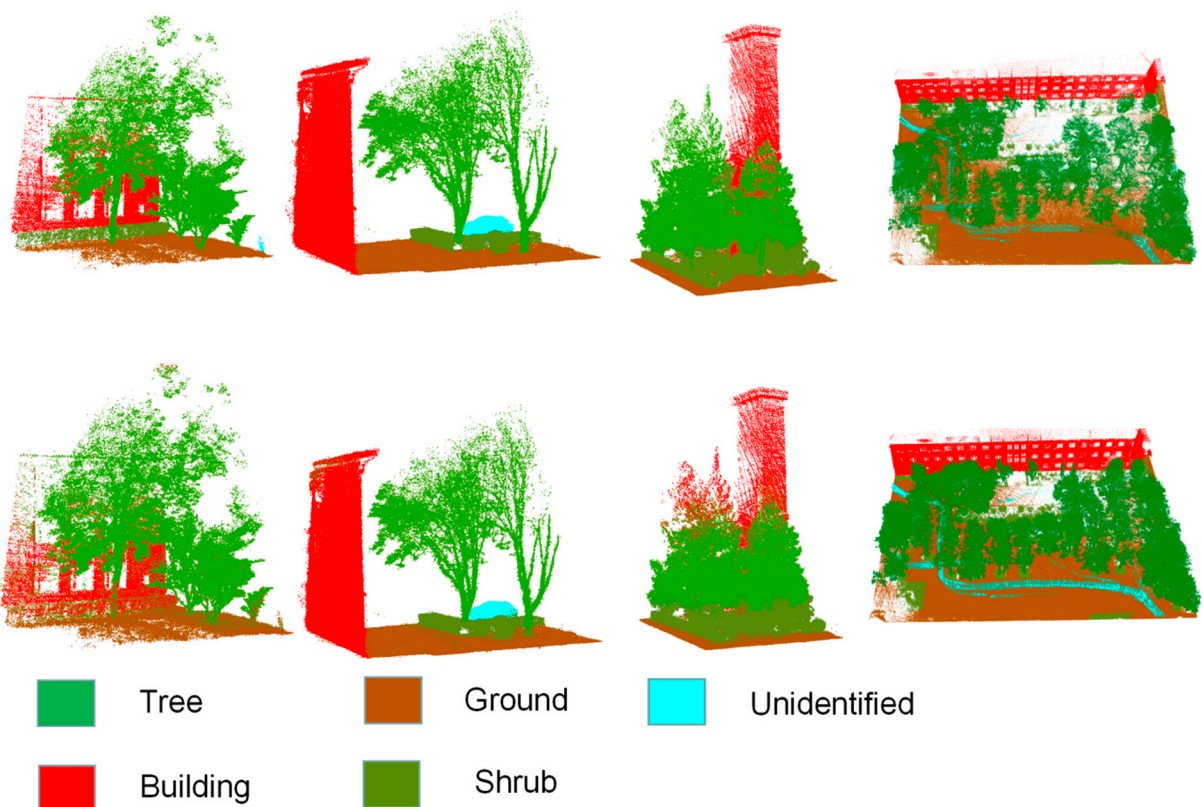

**Figure 11.** Example of PointDMS semantic segmentation visualization of the DMS dataset's scene c, semantic segmentation effect of Beijing Forestry University campus garden scene. Labels represent 5 different classifications. The first line shows true labels of the point cloud data. The second line shows visualizations of semantic information predictions obtained from the point cloud data.

We trained the DMS dataset's scene c using four methods, iterating 500 epochs, and achieved a final loss rate of 0.14. The setting of the learning rate is crucial to the learning process. A very low learning rate can result in slow learning speed, while a very high learning rate can make convergence difficult. Therefore, dynamic adjustment of the learning rate is often employed. Initially, a higher learning rate is set to accelerate the learning speed. Then, the learning rate is gradually reduced to search for the optimal solution. For this purpose, we used the Adam optimizer mechanism in PyTorch. The parameter configuration for Adam is as follows: alpha, also known as the learning rate or step size factor, controls the update rate of the weights (0.001). A larger value results in faster initial learning before the learning rate update, while a smaller value (0.001) allows for better convergence. Beta1 represents the exponential decay rate for the first moment estimate (0.9), and beta2 represents the exponential decay rate for the second moment estimate. This hyperparameter is set to a value close to 1 (such as 0.999) for sparse gradients (semantic segmentation of point clouds). Epsilon is a very small number (0.00001) used to prevent division by zero in the implementation. The parameter settings are as follows: alpha = 0.001, beta1 = 0.9, beta2 = 0.999, and epsilon = 0.00001. The training process utilizes the hardware and configuration mentioned in the previous subsection. The accuracy evaluation of the training results is presented in Table 5. We employed the evaluation metrics from the previous section to assess the training results. The overall accuracy (OA) reached 93.3, mIoU was 70.7, and there was a 12% improvement in building and other aspects.

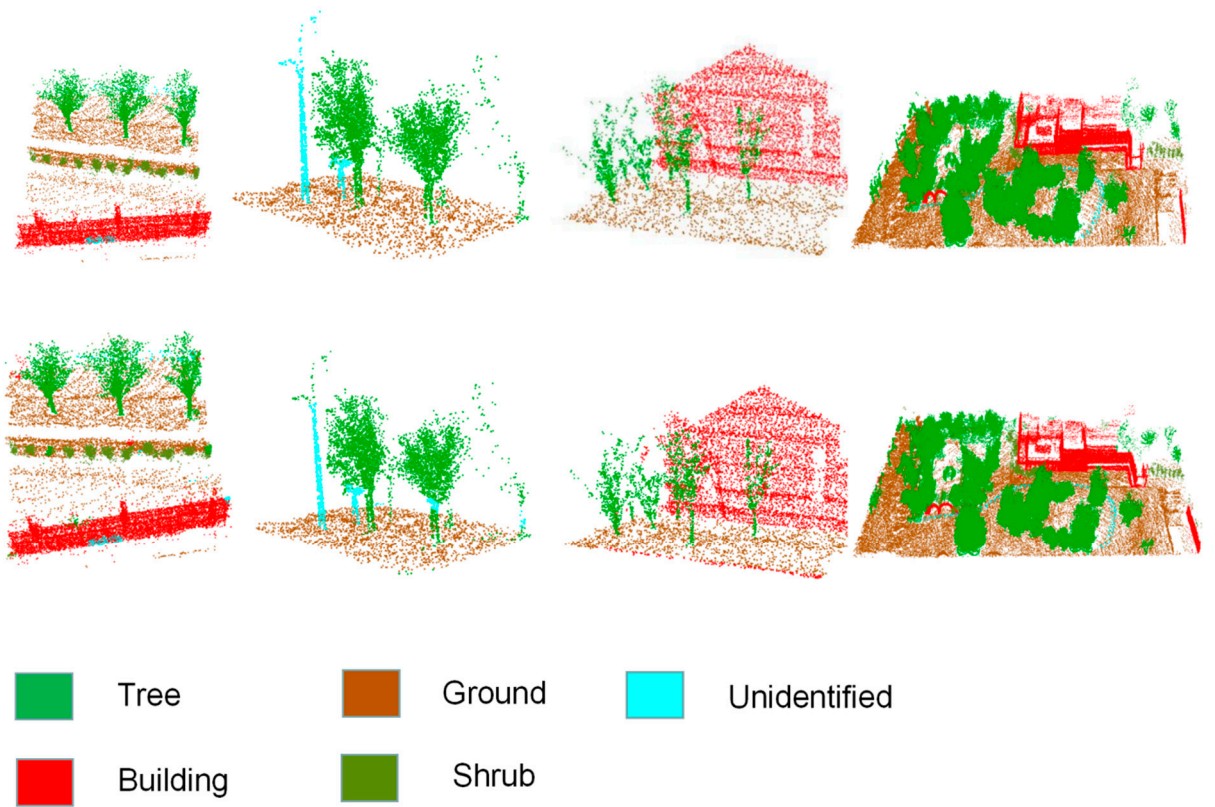

**Figure 12.** Example of PointDMS semantic segmentation visualization of the DMS dataset's scene b. Visualization of labels and predictions for Beijing World Park, with labels representing 5 different classifications. The first line displays true labels of the point cloud data. The second line shows visualizations of semantic information predictions obtained from the point cloud data.

**Table 5.** Accuracy of different methods and epochs in different scenes of high vegetation, low vegetation, and other targets.

| Accuracy | | Method | | | | | | |
|---|---|---|---|---|---|---|---|---|
| | | PointNet++ [13] | PointNet [12] | PointLae [40] | Spg [39] | Kpconv [21] | PointCNN [17] | PointDMS |
| 500 epochs in scene c | Ground | 78.1% | 74.5% | 73.2% | 81.5% | 92.7% | 90.3% | 94.5% |
| | Tree | 64.3% | 59.8% | 79.4% | 77.5% | 81.4% | 80.7% | 93.5% |
| | Shrub | 51.7% | 60.8% | 73.9% | 55.3% | 70.5% | 76.8% | 91.5% |
| | Building | 75.9% | 81.7% | 81.5% | 90.3% | 91.5% | 92.7% | 93.7% |
| | OA | 67.5% | 69.2% | 77.0% | 74.5% | 84.0% | 85.1% | 93.3% |
| 1000 epochs in scene b | Ground | 78.1% | 74.5% | 73.2% | 75.0% | 92.0% | 90.0% | 80.1% |
| | Tree | 64.3% | 59.8% | 79.4% | 78.0% | 65.0% | 79.0% | 87.5% |
| | Shrub | 51.7% | 60.8% | 73.5% | 71.0% | 68.0% | 70.0% | 77.5% |
| | Building | 75.9% | 81.7% | 81.6% | 94.0% | 96.0% | 95.0% | 92.1% |
| | OA | 67.5% | 69.2% | 76.9% | 79.5% | 80.3% | 83.8% | 84.3% |

We conducted training on the DMS dataset's scene b using seven different methods, with iterations over 1000 epochs. The final loss rate was reduced to 0.23. Table 5 presents the evaluation of the training results in terms of accuracy. We utilized the evaluation metrics discussed in the previous section to assess the training outcomes. The accuracy achieved was 84.3, with a mIoU of 74.3. Notably, the accuracy of tree recognition improved by 8%.

### 3.4. Validity Analysis of Living Tree Identification

Previous projects conducted by our team have primarily centered on enhancing the efficacy of point clouds for tree identification. Building upon this foundation, our current study investigates the efficacy of recognizing standing trees across various datasets. The outcomes of our analysis are presented in Figure 13.

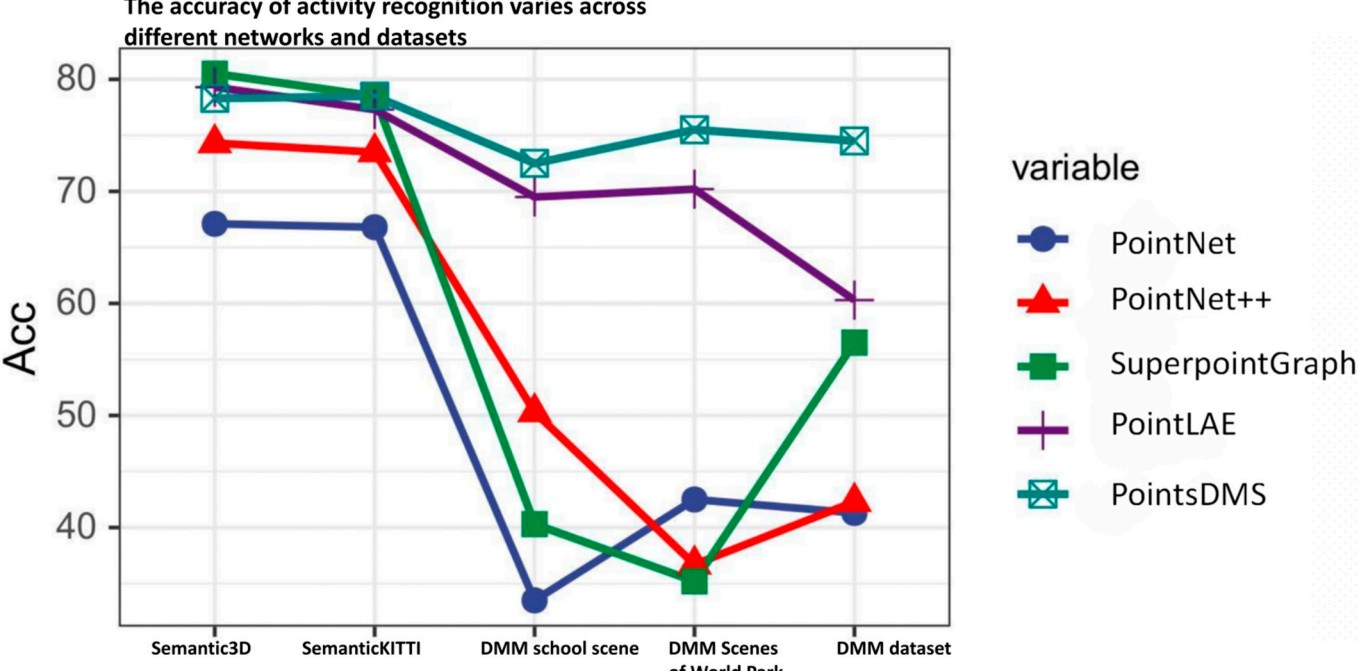

**Figure 13.** Effectiveness of different algorithms for forest tree identification under random recall.

We trained using a random drop rate of 0.3, resulting in a smaller number of tree samples in the training set. Additionally, we performed mixing of point clouds with varying sparsity. When comparing the accuracy of different algorithms applied to different datasets, our algorithm consistently achieves superior results. Specifically, the recognition accuracy for forest trees alone is improved by 14%.

## 4. Discussion

### 4.1. Evaluating PointDMS

In the field of traditional urban forestry, the primary method of data acquisition is still the traditional sample plot survey. However, the use of LiDAR scanning measurement can greatly enhance the efficiency of urban forestry information acquisition and facilitate the development of intelligent urban forestry. This paper proposes a method that effectively extracts information related to various vegetation structures from point cloud data obtained through mobile surveys in urban gardens. By doing so, a three-dimensional digital representation of urban gardens can be constructed, thereby promoting the development of intelligent gardens.

In this study, we employ a combination of computed point cloud geometric features and the XGboost method to obtain clusters of point cloud geometric features. These features are then merged and spread to a higher-dimensional space. Additionally, an improved over-segmentation method is applied to pre-segment the point cloud data. Our PointDMS algorithm effectively filters and outputs improved results.

Although our work represents significant progress in this field, there are still limitations to consider. Firstly, noise is inevitably generated during the data acquisition process due to equipment limitations. Secondly, manually labeling the point cloud data can be challenging due to its sparse, inhomogeneous, and complex nature, which may lead to misjudgments. However, we made every effort to minimize the impact of noise. In Figure 3, we have labeled the point clouds from different angles and with different sparsity levels and shading sizes.

Furthermore, by analyzing the segmentation results shown in Figures 11 and 12, it is evident that the segmentation accuracy may be compromised in scenes with uneven sparsity. We attribute this to the inherent errors that can occur during the sampling process in sparse scenes.

In summary, the segmentation method proposed in this paper for living trees in garden environments has proven to be effective. The model has been synthesized and trained using different scenes, and it has demonstrated excellent performance in our final test. Throughout the training process, we fine-tuned the parameters to obtain optimal weights, resulting in a model that is particularly effective in labeling trunk and leaf point clouds. Overall, the final model is robust and reliable.

### 4.2. Comparison with Similar Methodologies

Compared to other algorithms, our algorithm demonstrates robustness on the public dataset investigated. Both PointNet and SuperPointGraph share certain similarities with our structure. However, PointNet struggles to capture global features of point clouds, while SuperPointGraph addresses the limitation of point cloud input size. Nonetheless, SuperPointGraph's network structure is complex and computationally intensive, resulting in poor performance on irregularly shaped objects such as living trees. In contrast, our algorithm allows for unrestricted input and effectively extracts point cloud features, incorporating pre-segmentation to form multiple point cloud neighborhoods with similar features. This approach saves computation time and cost. As shown in Table 5, there is a significant improvement in the recognition accuracy for objects like trees. The segmentation of outdoor point clouds has garnered considerable attention in recent years. We selected several methods similar to those presented in Table 5. Although the scenes and point cloud sizes may differ, studies related to the selected methods focus on large-scale outdoor scene segmentation based on point clouds, providing valuable insights for our own research.

Among these studies, the authors of study [13], study [12], and study [40] demonstrate the overall accuracy of their work, with their methods performing well. However, our method holds a relative advantage in large scenes. Specifically, regarding the segmentation of living trees, our work is optimized to capture the distinctive features of living trees, enabling the neural network to recognize these features more effectively. In Figure 13, ref. [40] exhibits superior overall performance, but our method achieves a higher level of recognition for different living trees across all datasets.

### 4.3. Future Work

In the subsequent phase of the project, our focus will be on conducting long-term multi-sensor-fusion monitoring of standing trees in landscape environments. To expand our existing dataset, we will employ versatile tools such as drones to collect standing tree data from ALS gardens. We have already gathered raw data from drones and mobile cameras in various park test plots, encompassing a diverse range of standing trees. Furthermore, we aim to enhance the current methodology by exploring the integration of image data for flexible training. Simultaneously, we will refine the existing algorithm and streamline the neural network to optimize computational resources. This academic approach will contribute to the advancement of our research.

## 5. Conclusions

This paper presents PointDMS, a framework designed for the semantic segmentation of point clouds in urban forest environments. PointDMS utilizes feature aggregation and energy optimization techniques to effectively process, segment, and extract information from 3D point clouds. The framework introduces the DMS dataset, which facilitates urban environment recognition, and proposes a DMS module for efficient pre-processing of point clouds to extract enhanced geometric and global features. PointDMS achieves promising recognition results, demonstrating an overall accuracy of 93.3% and a recognition rate of 93.5% for live trees when trained on the DMS dataset. The robustness of PointDMS is further validated on the semantic3D dataset, where it achieves an overall accuracy of 88.3% and a recognition accuracy of 87.5% for standing trees. Notably, our algorithm surpasses other point cloud methods in terms of suitability for large and complex scenes, improving the recognition accuracy of standing trees by 8.2%. Looking ahead, our future work aims to optimize the neural network structure to achieve even better semantic segmentation results. Additionally, we plan to enhance the efficiency of the proposed algorithm to enable real-time segmentation applications in complex environments.

**Author Contributions:** Conceptualization, J.L. (Jiang Li) and J.L. (Jinhao Liu); methodology, J.L. (Jiang Li); software, J.L. (Jiang Li); validation, J.L. (Jiang Li); formal analysis, J.L. (Jiang Li); investigation, J.L. (Jiang Li); resources, J.L. (Jinhao Liu); data curation, J.L. (Jinhao Liu); writing—review and editing, J.L. (Jiang Li); visualization, J.L. (Jiang Li); supervision, J.L. (Jinhao Liu); project administration, J.L. (Jinhao Liu). All authors have read and agreed to the published version of the manuscript.

**Funding:** This research was funded by the National Natural Science Foundation of China (No. 62006008, 62173007, 62203020), the National Key Research and Development Program of China (No. 2021YFD2100605), and the MOE (Ministry of Education in China) Project of Humanities and Social Sciences (No. 22YJCZH006).

**Data Availability Statement:** Test methods and data are available from the authors upon request.

**Conflicts of Interest:** The authors declare no conflict of interest.

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
