# Peer review of "PointDMS: An Improved Deep Learning Neural Network via Multi-Feature Aggregation for Large-Scale Point Cloud Segmentation in Smart Applications of Urban Forestry Management"

_forests, doi:10.3390/f14112169_

Round 1

Reviewer 1 Report

In paper authors proposed the point cloud deep learning framework PointDMS: it consists of a multifeature set preprocessing segmentation module DMS and a deep learning network PointDMS, with the following innovations:

1) Built multi-sensor acquisition system and data processing process to collect point cloud data under different terrain environments,

(2) Proposed a so-called DMS module aiming at fast pre-processing, pre-segmentation, and maximize the retention of geometric features of point clouds for complex and large scale point cloud data. 

(3) Proposed a deep neural network PointDMS, which can be trained to segment and recognize the point clouds preprocessed by the DMS module effectively.

Comments:

1. line 91-127 - I do not like this text.The bullet point is without any prior signal the ":" character. This text seems ill-thought-out and too general. It should be changed. If references to literature are provided, I ask the authors to specify what these articles concern and what was checked/proposed in them, and what conclusions are drawn from these works in the context of the proposed solution.

2. page 3 line 130: built not build

3.page 4 line 155:  segmented and filtered using Geomagic software. 

What does segmentation and filtration mean for authors? they seem to be confusing these terms.

4. page 4 line 156: I don't understand this sentence:  "And the individual targets are sampled after mirroring to obtain sparse point cloud data". 

5. Figure 1:  if there are 4 different sources of obtaining clouds, where is the registration of these clouds in the diagram? there is no stage of preparing the point cloud for segmentation.  After the steps mentioned in pre-processing, it does not follow that we will obtain classified point clouds.

6. What sampling/reduction/optimization method was used?

7. pge 7 line 211-224: this text doesn't fit here. Rather, it needs to be moved somewhere to be implemented. This is basic information about measuring with a laser scanner.

8. Figure 4 and Figure 5: Figures are not very legible, there is too much information on these diagrams. you should simplify it and definitely not add formulas to the diagrams. What do the abbreviations in the MLP diagram mean? and others, it should all be explained. ATSE moduel? KPCONV?

9. Table 4. Accuracy with different method and epochs in different scenes. what units are these? is this the percentage of detection?

This paper is poorly written, it is difficult to complete the review. There are no clearly described stages.

The goal seems good, but the work itself, in my opinion, does not meet the requirements of scientific work. For results to be reliable, they must be clearly presented. What is missing here is proper paper management.

Reviewer 2 Report

Deep learning for 3D semantic segmentation in forestry point clouds is a popular topic and is worth studying. The authors propose a semantic segmentation framework as well as providing a new dataset that they collected. The results show that their accuracy has better results. 

However, the manuscript has plenty of reference typos and clarification issues and requires edits to improve the readability and representation. Particularly, providing more data for efficiency instead of only accuracy and emphasizing the differences from the recent literature will enhance the quality of the paper. The authors should consider the comments and answer the questions given below;

Comments:

1)     The title may require a reconsideration. A colon may make the title more clear, as “.... Understory Environments: PointDMS”

2)     A revisit for the references is required, both in the text and reference section.

3)     Line 96-127 – Revisiting existing models required. The text includes duplicates [Line117-122]. The categorization of the methods is unclear as well as some papers are not correctly grouped, such as KPConv is a convolution-based method, while point web is an extension to PointNet++ and should be in the group of point-based methods [In the literature, Pointnet++ is a point-based method, but the technique is a graph-based method, aggregating the features from the neighbors on radius graph or knn graph. Graph convolutional neural networks mainly focus on generated graphs from the point cloud data].

A review paper may help to revisit this section, as well as for clarification of the methods.

Example paper:

Guo et al., 2020. Deep Learning for 3D Point Clouds: A Survey. IEEE TPAMI. doi: 10.1109/TPAMI.2020.3005434

4)     Lines 137-147: The innovations require a revisit, details and advantages may come forward, instead of general benefits of both the deep learning framework and DMS module.

In addition, naming the deep learning framework PointDMS and preprocessing module as DMS creates confusion; instead, they can be renamed for fluency.

5)     Table 1 font size is larger, and the font is different than the default font.

6)     Table 2 heading font size is different from the table

7)     Line 269-270. The authors say that the part of the dataset is open to the public through the Baidu cloud disk. Is it more convenient to provide a link for the download, if it is not breaking the licensing issues for the dataset? Also, may provide a note in acknowledgment, of how to reach the dataset.

Also, are there any similar datasets in the literature? If there, what are the similarities and dissimilarities to your dataset? Moreover, as a suggestion, it is going to be great to see the number of samples per class as a table or a figure to have a summary at the end of the dataset section.

8)     In line 151, “.. the point cloud multi-feature pre-segmentation module DMS module...”, in line 271, “2.2 PointDMS fusion module”. Are those the same? Is the DMS module also renamed as the PointDMS fusion module? A clarification is needed at this point.

9)     Line 291. What is the ATSE module? It seems in Figure 4 that it is a part of the DMS module, and touched in Lines 295-299; however, it is unclear.

Also in Lines 295-298, the first ATSE method used than Xgboost used for valid features. But in Figure 4, the order is different. A clarification is needed.

In Figure 4, geometric features and intensity features are processed through MLP and concat. Why the authors process those Nx3 and Nx1 data in a separate MLP then concat should require clarification. What is the benefit of processing Nx4 in one MLP? Figure 4 needs modification for the sake of understanding, and also combination with the text is essential for a solid explanation.

What is V in Eq.4? Formulas in Eq. 4,5,6 and 7 need clarification.

In Figure 4 and Line 296, is Xgboost used, with default parameters, or is parameter tuning performed?

The authors also should cite Xgboost's paper.

There is ATSE v2 in Figure 4, what is it?

Line 311, what is L&O? Revisit for citations required.

10)  Figure 5 needs clarification.  In the caption, the authors say that the framework includes 3 main parts, but defines the left part and middle part, no right part is clearly defined. Also, in the middle part of the figure, both up and down networks are identical?

In Figure 5, Tnet proposed, which is a part of Pointnet, not referred to in the text.

11)  Lines 352-367 require clarification.

12)  Algorithm 1 requires clarification. Python style indentation and explicit input and output may improve the representation.

13)  Is the PointDMS framework an end-to-end framework? If it is, it should improve the paper to imply what is the input for the framework.

14)  Figure 6 caption includes training accuracy and loss curves, but in Lines 378-379, it says “...validation accuracy and training loss obtained ....”. Needs clarification for both the figure and text.

15)  If validation data is used, please explicitly define it in the text, in addition to comment 14.

16)  Line 430, what is the optimal learning rate? Is it tuned or empirically determined?

17)  In Figure 11, up is the true, and down is the predicted? Same for Figure 12? Write them in the captions of the Figures.

18)  Table 4, it is better to make the bold font the best number. It improves representation and understanding.

19)  Referring to Lines 522-532, what is the most similar framework to your proposed framework in the literature? SuperpointGraph? In line 527, the authors state that in large areas their method is advantageous, what are those? The number of parameters, the inference time, and the training time? What makes the framework advantageous for large scales?

20)  Line 468. Table 2 should be Table 4.

21)  Line 525 Table 2 should be Table 4.

22)  Line 524-525. Under which settings were used for the DL models in Table 4? Default settings from the original papers?

In Table 4, PointNet++ is in the MultiScaleAggreation mode?

23)  What don’t have results for SPG, KPCONV, and PointCNN for 500 epochs in scene c in Table 4?

Reviewer 3 Report

COMMENTS TO THE AUTHORS:

- in the keywords Section, I don't understand the the inclusion of the last item "Multiple feature aggregation".  It is not very significant.

- line 95: it should end in ":" so that the following paragraphs hang from it.

- line 139: it should end in ":" so that the following paragraphs hang from it.

- line 156: there is no reference to the GEOMAGIC Software.

- Empty lines between 181 and 208 should be eliminated.

- Table 1 is not referenced in the text (at least I can't find it).

- line 225: I would describe the different equipment in independent numbered paragraphs.

- line 269-270: no website is given for downloading the data.

- Perhaps, equations (1), (2) and (3) in lines 279, 280, should be located after Figure 4.

- Lines 363-364: it is said that figure 6 shows the semantic segmentation by interpolation method. However, figure 6 refers to the "Training accuracy curve and loss curve of PointDMS".   Maybe the authors should rename the title of figure 6.

- line 482: citations [36-44] are inserted, but these citations do not exist in the References Section. 

- In the References, the numbers appear twice: Example 1. 1.Nelson R, Krabill W,...

Round 2

Reviewer 1 Report

Thank you for taking all comments. The work gained scientific value.

 At the same time, I think that each existing method used in the paper should be explained not only by reference to literature. Please remember this for the future.

Author Response

Responds to the reviewers’ comments

Dear Editors and Reviewers:

Thank you for your letter and for the reviewers’ comments concerning our manuscript entitled (forests-2630485). Those comments are all valuable and very helpful for revising and improving our paper, as well as the important guiding significance to our researches. We have studied comments carefully and have made a correction which we hope meets with approval. We tried our best to improve the manuscript and made some changes to the manuscript according to your suggestions. These changes are marked in highlighted style, which will not influence the content and framework of the manuscript.

We appreciate for editors' and reviewers’ warm work earnestly and hope that the correction will meet with approval. The main corrections in the paper and the responses to the reviewers’ comments are as following:

Reviewer #1:

General comments:

Thank you for taking all comments. The work gained scientific value.

 At the same time, I think that each existing method used in the paper should be explained not only by reference to literature. Please remember this for the future.

Response: We sincerely appreciate your positive feedback on the manuscript. We are honored that our method has sparked your interest. Your positive comments serve as great encouragement for us. Additionally, we would like to express our gratitude for bringing to our attention the errors in our paper, as this feedback is highly valuable to us.In the new round of revisions, I have highlighted the modified parts.

Reviewer #1:

General comments:

The authors improved the paper with additional information and replies to the questions. But still, there are parts that need to be clarified;

Response: We greatly appreciate your positive comments about the manuscript. We are honored that our method arouses your interest. The positive comments you made are the greatest encouragement to us. Thanks for pointing out some grammatical errors in our paper, which is very useful for us.In the new round of revisions, I have highlighted the modified parts

Detailed comments:

1)      (Comment 2 in previous version)

References in the text still have problems such as:

Line 64 -. Schreier et al. …?

Line 70 - … Merlijn Simonse [3] et al. ..

Line 71 - …Aleksey Golovinskiy [4] et al….  

Please check the format of the journal.

 Response:Thank you very much for your valuable suggestions. We have re-audited the document format and made modifications. The new manuscript highlights lines 65 to 73.

2)      (Comment 3 in the previous version)

Existing methods in the text have been improved, thanks to this, but it still needs revisiting:

Line 98 – Traditional voxel ...

Line 108-112 – Do they belong to PointMLP methods? What about shape classification?

 Response:Thank you very much for your valuable suggestions. I have reorganized and highlighted this part of the content. Please refer to the new manuscript from line 99 to line 113.

3)      (Comment 7 in previous version)

  1. A) Thanks for the existing datasets. In table 3, what is the dataset name? DMM or DMS? Only once I have encountered DMM, it is in the table caption, then in the text, it refers to the DMS dataset, such as:
  2. Line 195 – DMS dataset …
  3. B) Another point is that assuming your dataset is DMM, the number of points in the table and the text does not match.

Line 304-305: “After the initial sampling process, there are still more than 19 million points in the whole point cloud datasets.”

Similar to scene numbers, in the text, Line 327, it is 12530, but in the table, it is 13680.

So, you have 19 million points or (according to the table) approximately 1 300 million points?

  1. C) Which one is the most similar to your dataset? Is your dataset like SemanticKitt, where each scene almost has 1 million points, or Paris Lille or Semantic3D, processed and created a huge dataset?

Clarification and similarities need to be identified.

  1. What about Bjfumap and bfumap mix labels in Figure 13? Are those datasets? If there are separate datasets available, referring to those in the dataset table would improve the paper.

Response:Thank you very much for your valuable suggestions. We apologize for the mistake with DMM, which was a typographical error on our part. We have reviewed the entire document and made corrections in Table 3. Regarding B, we have reevaluated our work and made corrections to the original manuscript. The DMS dataset includes 1150 scenes in campus environments and 12530 scenes in park environments, totaling 13680 scenes. We have made revisions and highlighted the explanations from lines 340 to 344 in the text.

Figure 3 shows samples of our collected DMS dataset tag visualization, which contains 1150 small scenes from the campus of Beijing Forestry University and the World Park scenes incloude 12530 environmental scenes,Our DMS dataset has a total of 13,680 scenes.Specific details are shown in Table 3 and compared with other point cloud datasets.

Regarding the issue of points, we have also made adjustments, highlighted in lines 310 to 322.

After the initial sampling process, there  still more than 1299 million points in the whole point cloud datasets.

For C, our dataset is similar to semantic3D, and we have added relevant introductions to it, from line 331 to 352, with highlighted markings.

For D, I apologize for the naming error with Bjfumap and bfumap. They are actually our DMS dataset's campus and park scenes. I have made modifications and highlighted them in Figure 13. Please refer to line 644 for more details.

4)      (Comment 8 in previous version)

What is BFU in the subtitle of Chapter 2.2? Write the open version and explain it in the paper.

 Response:Thank you again for your valuable suggestions. I apologize for the previous mistake in my work. I have now made the necessary changes to the title in section 2.2. It is highlighted on line 358.

5)      (Comment 16 in the previous version)

  1. A) Line 600-605, Line 560-563: Which learning rate scheduler is used with which optimizer? A clear indication improves the quality. Reporting those will help to reproduce results. Also, In Line 563, the text referring to Figure 7, should be Table 4.
  2. B) What are the “Study attrition rate” and “Base study rate”? From my understanding, they are learning rate and learning rate decay? A clear indication of training parameters may improve the results. Initial learning rate, learning rate decay, scheduler, optimizer, etc. details may improve the representation as well as allow to reproduction of the results.

 Response:Thank you again for your valuable suggestions. In this article, we used Adam for optimization. We have updated Table 4 and added descriptions of relevant optimizer details. We described the basis for using the Adam optimizer from line 590 to 595 and explained the Adam optimizer parameter settings for our training process from line 636 to 648. Finally, we correctly referenced Table 4 at line 595 and highlighted it in the updated manuscript.

6)      (Comment 19-20-21-22-23 in the previous version)

The paper uses Precision, Recall, F1, and OA for the metrics. But in Lines 611 and 620, the authors report with mean Intersection with Union. A clarification of metrics is also needed for comparison with the existing methods. 

 Response:

Thank you very much for your valuable suggestions. The introduction of evaluation indicators such as Miou has been updated in the new manuscript, specifically in lines 519 to 534, and highlighted.

Reviewer 2 Report

The authors improved the paper with additional information and replies to the questions. But still, there are parts that need to be clarified;

1)      (Comment 2 in previous version)

References in the text still have problems such as:

Line 64 -. Schreier et al. …?

Line 70 - … Merlijn Simonse [3] et al. ..

Line 71 - …Aleksey Golovinskiy [4] et al….  

Please check the format of the journal.

2)      (Comment 3 in the previous version)

Existing methods in the text have been improved, thanks to this, but it still needs revisiting:

Line 98 – Traditional voxel ...

Line 108-112 – Do they belong to PointMLP methods? What about shape classification?

3)      (Comment 7 in previous version)

A) Thanks for the existing datasets. In table 3, what is the dataset name? DMM or DMS? Only once I have encountered DMM, it is in the table caption, then in the text, it refers to the DMS dataset, such as:

a.       Line 195 – DMS dataset …

B) Another point is that assuming your dataset is DMM, the number of points in the table and the text does not match.

Line 304-305: “After the initial sampling process, there are still more than 19 million points in the whole point cloud datasets.”

Similar to scene numbers, in the text, Line 327, it is 12530, but in the table, it is 13680.

So, you have 19 million points or (according to the table) approximately 1 300 million points?

C) Which one is the most similar to your dataset? Is your dataset like SemanticKitt, where each scene almost has 1 million points, or Paris Lille or Semantic3D, processed and created a huge dataset?

Clarification and similarities need to be identified.

D) What about Bjfumap and bfumap mix labels in Figure 13? Are those datasets? If there are separate datasets available, referring to those in the dataset table would improve the paper.

4)      (Comment 8 in previous version)

What is BFU in the subtitle of Chapter 2.2? Write the open version and explain it in the paper.

5)      (Comment 16 in the previous version)

A) Line 600-605, Line 560-563: Which learning rate scheduler is used with which optimizer? A clear indication improves the quality. Reporting those will help to reproduce results. Also, In Line 563, the text referring to Figure 7, should be Table 4.

B) What are the “Study attrition rate” and “Base study rate”? From my understanding, they are learning rate and learning rate decay? A clear indication of training parameters may improve the results. Initial learning rate, learning rate decay, scheduler, optimizer, etc. details may improve the representation as well as allow to reproduction of the results.

6)      (Comment 19-20-21-22-23 in the previous version)

The paper uses Precision, Recall, F1, and OA for the metrics. But in Lines 611 and 620, the authors report with mean Intersection with Union. A clarification of metrics is also needed for comparison with the existing methods. 

Author Response

(The authors gave the same response as above.)
